# Moisture Sorption and Degradation of Polymer Filaments Used in 3D Printing

**DOI:** 10.3390/polym15122600

**Published:** 2023-06-07

**Authors:** Andrey Aniskevich, Olga Bulderberga, Leons Stankevics

**Affiliations:** Institute for Mechanics of Materials, The University of Latvia, Jelgavas Str. 3, LV-1004 Riga, Latvia; olga.bulderberga@lu.lv (O.B.); leons.stankevics@lu.lv (L.S.)

**Keywords:** polymer, filament, moisture, sorption, Fick’s law, cylinder, 3D printing, swelling, elastic modulus, strength

## Abstract

Experimental research of the moisture sorption process of 12 typical filaments used for FFF was performed in atmospheres with a relative humidity from 16 to 97% at room temperature. Materials with high moisture sorption capacity were revealed. Fick’s diffusion model was applied to all tested materials, and a set of sorption parameters was found. The solution of Fick’s second equation for the two-dimensional cylinder was obtained in series form. Moisture sorption isotherms were obtained and classified. Moisture diffusivity dependence on relative humidity was evaluated. The diffusion coefficient was independent of the relative humidity of the atmosphere for six materials. It essentially decreased for four materials and grew for the other two. Swelling strain changed linearly with the moisture content of the materials and reached up to 0.5% for some of them. The degree of degradation of the elastic modulus and the strength of the filaments due to moisture absorption were estimated. All tested materials were classified as having a low (changes ca. 2–4% or less), moderate (5–9%), or high sensitivity to water (more than 10%) by their reduction in mechanical properties. This reduction in stiffness and strength with absorbed moisture should be considered for responsible applications.

## 1. Introduction

Additive manufacturing, widely known as 3D printing, is a modern technology that has experienced constant growth in recent decades. It is also referred to as one of the trends of Industry 4.0 [1]. One of the AM technologies, known as fused filament fabrication (FFF), has great potential for engineers and designers and allows the creation of unique complex-shape polymer constructions within relative time limits [2,3]. FFF technology is based on a layer-by-layer layup of extruded polymer material, referred to as filament. The filament materials range from pure polymers to those reinforced with nanoparticles, fibres, etc., responding to planned applications [4]. Generally, polymer materials used as a filament base can be divided into home/laboratory use and industrial use [5]. Materials for home/laboratory use are more straightforward in terms of printing and processing but are more sensitive to environmental factors [6] and do not have outstanding mechanical and physical properties [7]. Such materials could be used for prototyping and producing parts for single use and for studying the material properties [8]. The main materials used in this group are polylactic acid (PLA), acrylonitrile butadiene styrene (ABS), polycarbonate (PC), and polyamide (PA, Nylon). On the other hand, materials for industrial printing such as polyetherimide (PEI) and polyetherketoneketone (PEKK) materials, or a combination of them [9,10], have more advanced properties, are not so sensitive to external factors, and could be used for more practical applications [11].

Three-dimensional printing using the FFF method is swiftly moving from prototyping to functional parts, thus facing new practical challenges. Most traditional polymers and composites used for structural applications are moisture-sensitive, tend to absorb humidity, and lose stiffness and strength in a moist environment [12,13]. FFF filaments are expected to behave similarly. Thus, this affects the printing parameters [6] and properties of the 3D-printed structures [14,15].

Filaments shipped from manufacturers are vacuum-packed and are in relatively stable conditions before opening. The mechanical, physical, and other properties mentioned in the datasheet should correspond to these reference, “ideal” conditions. After unsealing, filaments tend to sorb the moisture from the ambient atmosphere, which is typically not controlled. Frequently, only specialised 3D printing companies closely monitor the storage conditions of unsealed filaments during the storing and printing period. For example, such companies use a 3D printing filament storage cabinet or at least a silica-gel-containing storage box. In most room conditions, filaments are subjected to an atmosphere of 20–50% relative humidity (RH). Filaments absorb water from the air, swell, and degrade in such conditions. Even following the rules during storage for filaments, a coil is removed from the safety conditions, and the filament is subjected to humidity because most desktop-type printers are not equipped with a specific “humidity-safe” place or dry box for coil storage during the printing period.

Despite the wide range of work devoted to studying the properties of printed parts and printing parameters, a gap in the information is observed when it comes to studies of the main component—a filament.

One of the possible ways to investigate the effect of moisture on the mechanical properties of FFF structures is the so-called structural approach, well known in the mechanics of fibre-reinforced composites and also applied to moisture sorption processes [16]. In this way, the experimental investigation and modelling of filament moisture sorption are crucial. The sorption of various materials such as PLA, SiC-filled PLA composites, ABS [17], PC/ABS blend, and Nylon [18] was studied. The results showed that the sorption of samples was strongly affected by the manufacturing method [19,20]. Thus, these data are not applicable for sorption modelling in contrast to the sorption data of specific filaments. Unfortunately, systematised comparative research devoted to filaments is missing in the literature.

On the other hand, the moisture sorption process and its effect on the mechanical properties of composites are well investigated. The effect of moisture on the elastic and viscoelastic properties of epoxy and epoxy-based carbon-fibre-reinforced plastic filled with multiwall carbon nanotubes was described in [21]. The effects of moisture on the elastic and viscoelastic properties of CFRP rebars and vinyl ester binder are described in [22]. The hydrothermal ageing of an epoxy resin filled with carbon nanofillers is described in [23]. These attempts should be extended to FFF materials and structures.

This research aimed to conduct an experimental investigation and model the water sorption process of typical polymer filaments used for FFF in order to obtain the data necessary for the further prediction of the effect of moisture on the properties of FFF structures. To achieve this aim, four tasks were formulated: (1) To perform experimental research on the moisture sorption process of some typical filaments used for FFF. (2) To model the moisture sorption process and obtain the material parameters that characterised the process. (3) To evaluate the swelling of the filaments in wet environments. (4) To estimate the possible degree of degradation of the mechanical properties of the filaments due to moisture absorption.

## 2. Moisture Sorption by a Cylindrical Specimen

Let us consider the well-known Fick’s second equation [24,25] for the case of the moisture diffusion (sorption) process in a cylindrical polymer specimen
(1)$$\frac{\partial U}{\partial t}=D\Delta U$$
where *U* is the moisture concentration in the specimen, *D* is the diffusion coefficient, and *t* is time. The cylindrical coordinates are *r* and *z* only. The initial moisture concentration distribution in the specimen is *φ* (*r, z*), and the concentration on the specimen surface is *U*_0_. Equation (1) with initial and border conditions could be rewritten as
(2)$$\frac{\partial U}{\partial t}=D\Delta_{r,z}U,\;U{|}_{t=0}=\varphi \left(r,z\right),\;U{|}_{r=R}=U_{0},\;U{|}_{z=0,l}=U_{0}$$

The above-mentioned classical books provide the solution to the diffusion problem for an infinite-length cylinder. Many authors often use it to model the moisture sorption process in long fibres [26,27]. For the case of a bounded cylinder, instead of (2), we have
(3)$$\frac{\partial U}{\partial t}=D\left(\frac{\partial^{2}U}{\partial r^{2}}+\frac{1}{r}\cdot \frac{\partial U}{\partial r}+\frac{\partial^{2}U}{\partial z^{2}}\right)$$
with the initial condition
(4)$${\left.U\right|}_{t=0}=\varphi \left(r,z\right)$$
and the border condition
$${\left.U\right|}_{z=0,l}=U_{0},\;{\left.U\right|}_{r=R}=U_{0},\;\forall r,z,t\;U<\infty .$$

A complete step-by-step solution of the problem using the method of the separation of variables is given in Appendix A.

As a result, the moisture content in a cylindrical sample could be calculated as
(5)$$Q=Q_{\infty }+\frac{8\left(Q_{0}-Q_{\infty }\right)}{\pi^{2}}\sum_{k=1}^{\infty }\sum_{m=1}^{\infty }\frac{\exp\left(-\lambda_{k,m}^{2}Dt\right)}{m^{2}\gamma_{k}{}^{2}}{\left(1-{\left(-1\right)}^{m}\right)}^{2},\;\lambda_{k,m}^{2}={\left(\frac{\gamma_{k}}{R}\right)}^{2}+{\left(\frac{\pi m}{l}\right)}^{2}$$

The roots, *γ_k_*, of the zero Bessel function are also given in Appendix A.

A user-defined function was developed in Microsoft Excel using Visual Basic for Applications (VBA) programming language to calculate moisture content in a cylindrical sample using (5).

A large number of summands (up to *k* = 30) in expression (5) provides convergence of the equation for relatively small time values close to zero, as seen in Figure 1. For the prediction of the moisture sorption process at the final stage, when the moisture content of a sample is 80% of the equilibrium or more, the sum could be restricted only to the first term of the series to obtain the simple expression
(6)$$Q=Q_{\infty }+\frac{32\left(Q_{0}-Q_{\infty }\right)\exp\left(-\lambda_{1,1}^{2}Dt\right)}{\pi^{2}\gamma_{1}{}^{2}}$$

This expression provides 98% or better accuracy for Fo > 0.004, which is enough for many evaluations.

The moisture sorption rate depends on the geometry of the sample. A calculated graph of moisture sorption for cylinders with different length-to-radius ratios *l*/*R* is presented in Figure 2. It is seen from the figure that for short cylinders with *l*/*R* < 2 (disk shape), the contribution of the axial component in the sorption process is essential. The one-dimensional solution with only the radial component gives an underestimated prediction for such cases. For modelling purposes, the equilibrium moisture content in this calculation was chosen and fixed as *w*_∞_ = 1, but this does not matter for such evaluation.

## 3. Materials and Methods

A list of tested materials is given in Table 1. Representatives of the most often used filament classes were tested. Some materials with additional functionality, such as electrical conductivity (Koltron, PLA Cnd, ABS ESD) or a thermochromatic effect (PLA LAVA), were also included in the list. The fillers that provide this additional functionality can notably influence the properties of basic polymers; however, these effects are beyond the scope of the present study. Most of the tested thermoplastics are hydrophobic materials, excluding Nylon and CPE, which are hydrophilic, as indicated in the technical data sheets. This information is not available for materials with additives.

The categories of tested materials include common, engineering, functional, and high-performance plastics, as presented in Figure 3. This division is often used but is relatively conventional, and white horizontal lines in the figure are not sharp borders between the categories. The white vertical line in the figure is also not a sharp border between amorphous and semi-crystalline. Additionally, the horizontal and vertical arrows do not indicate any degree of crystallinity or the scale of price or performance, but only growth directions. The 13 tested polymers have different chemical formulations and molecular and sub-molecular structures that could change with moisture sorption. Their moisture sorption behaviour and degradation of properties also are different, though they should be classified using the obtained results.

All filaments were stored in the producer’s vacuum package with silica gel to prevent filament exposure to humid air before the start of experiments. Two types of samples were prepared and tested: “short” cylinders of *l* = 4 mm marked with an asterisk (*), as given in Table 1, and “long” cylinders with a length of *l* = 100 mm (all materials are listed in the table, except ABS Black). The nominal diameter of the filaments was 2*R* = 2.8 mm (except Ultem and Antero, with a diameter of 2*R* = 1.75 mm). “Short” cylinders were intended for accelerated moisture sorption tests, while “long” cylinders, in turn, were for long-term sorption, swelling, and mechanical tests. All samples were cut with a knife and the cylinder ends were polished.

Moisture sorption experiments were performed in desiccators with varying levels of RH of the atmosphere created using silica gel (12%) and saturated salt solutions: LiCl (16%), KSCN (47%), NaCl (75%), and K_2_SO_4_ (97%). From 10 to 15 “short” and 5 “long” cylinder samples were placed in each desiccator. Silica gel was dried in an oven at a temperature of 150 °C for 3 h. A datalogger Extech Instrument RHT10 (Extech Instruments, NH, USA) was used to control the humidity, temperature, and dew-point values in the desiccators. It was revealed that after a desiccator lid was opened during the experiment, it took up to 20 h to restore the moisture environment in the desiccator after closure. The mass of the samples was controlled only once per day at the initial stage of the sorption process and later once per week using the Mettler-Toledo XS205DU Analytical Balance Scale (Mettler-Toledo, Switzerland) range 81/220 g with an accuracy of ± 0.00001 g. The relative water content of a sample *w*(*t*) was calculated using the equation
$$w(t)=\frac{m(t)-m_{0}}{m_{0}}\cdot 100\%$$
where *m*(*t*) and *m*_0_ are the current and initial mass of a sample.

The samples were considered to be in the equilibrium state if small mass oscillations were observed only around a value during a certain time interval, which was approximately the same as that of the sorption process. All moisture content values during the interval were averaged and considered as the equilibrium moisture content.

The length of “long” cylinders was measured for swelling calculations during moisture sorption experiments. A Mitutoyo Absolute digital micrometre 100–125 mm (Mitutoyo, Japan) with a precision of ± 0.001 mm was used for the measurements. This parameter has quite a big scatter in terms of its physical nature because the length of a sample changes by ca. 0.1% or less. Multiple length measurements for “long” samples were performed to reduce the data scatter.

Mechanical tensile tests of filaments moistened in a humid atmosphere till equilibrium were performed on the ZWICK 2.5 machine (ZwickRoell GmbH, Germany) with a 10 mm/min crosshead displacement rate. The tensile strain was measured by grip-to-grip separation with a nominal distance of 50 mm. A pre-load of 1 N was applied to all samples. The elastic modulus was calculated in the strain range of 0.05–0.25%. The tensile test duration till fracture depended on the material, but it was less than 1 min. Thus, the mass loss of absorbed moisture was neglected during the test. Three to five moistened specimens were tested for each material and each environment, and the average data were used.

## 4. Results and Discussion

### 4.1. Water Sorption

The sorption behaviour of all tested materials follows the classical Fick’s law. Figure 4 represents typical moisture sorption curves for Antero in atmospheres with various levels of RH. The sorption process had one stage, the sorption curves had initial linear segments, and the moisture content in samples reached equilibrium.

All filaments, as produced, had some unknown initial moisture content. The tested samples were not preliminarily dried before the sorption experiments. This fact explains why the filaments lost their mass in desiccators with a dry atmosphere. The ability of a material to absorb moisture could be characterised by the difference between the moisture content value obtained in the driest and wettest atmospheres. The moisture sorption capacity of the tested materials, Δ*w*, in 24 h is presented in Figure 5.

Materials with low (ca. 0.1% or less), moderate (0.1–0.5%), and high (more than 0.5%) moisture absorption capacity in 24 h could be classified based on the data in Figure 5.

Two unknown variables in (5) characterising moisture sorption kinetics, *w*_∞_ and *D*, were found using curve fitting on the experimental sorption data for all materials and all conditions. The aim function minimised during the fitting procedure was calculated as the average relative deviation. A set of equilibrium moisture content values for various values of RH give the sorption isotherm of a material. The linear approach to this dependence on RH is well known as Henry’s law. Nonlinear dependencies are known as Brunauer, Emmett, and Teller (also known as BET) classification. For practical applications, the following equation was used for both cases:(7)$$w_{\infty }=a\cdot {\mathrm{RH}}^{c}+b$$
where *a*, *b*, and *c* are coefficients unique to each material. Typical sorption isotherms of Ultem (linear, Henry’s law) and ABS White (BET, Type 3) are presented in Figure 6. In the figure, experimental dots are shown, their approximations given by solid lines and shifted approximations for the materials if they should be initially dry (dashed lines). This shift means that the parameter *b* = 0.

The coefficients of the approximation (7) and maximal (in equilibrium) moisture sorption capacity Δ*w_∞_* for all tested materials are presented in Table 2.

The isotherms for all materials are shifted down along the ordinate axis (parameter *b* is negative) because all of the tested filaments were not dry before the experiments but were used as produced by the manufacturers. The maximal moisture sorption capacity of the tested materials, Δ*w*_∞_, for the equilibrium state is also presented in Table 2.

Comparing the maximal moisture absorption capacity in 24 h (Figure 5) or at equilibrium (Table 2) for the tested materials, it could be concluded that this parameter does not depend significantly on the material type, whether the material is hydrophobic or hydrophilic, or the crystallinity. An exception is Nylon, with its extremely high absorption capacity and possible chemical changes in its structure, such as hydrolysis during absorption [29]. Please note that the density of Nylon presented in Table 1 is very low compared with other tested materials. Active fillers such as carbon black, often not disclosed by manufacturers, do not change the maximal moisture absorption capacity for PLA LAVA or PLA Cnd, while more than an order of magnitude increases this parameter for ABS ESD with respect to ABS White. The density of ABS ESD is the lowest for the tested materials. The low density of Nylon and ABS ESD likely means a high free volume of the materials that is occupied by absorbed water and that provides the high moisture sorption capacity of both materials. More detailed and complex research is required to reveal the connections between the structure of filaments and their moisture absorption behaviour, especially considering possible recrystallisation and density changes, e.g., for PLA filament and printed samples, as discussed in [28].

The second parameter found during the curve fitting of calculation (5) to experimental sorption curves was diffusivity, *D*. The coefficient could be assumed as independent of the moisture content for six tested materials. This independence was preliminarily supposed and expected according to Fick’s law. For the other six materials, the diffusion coefficient changed with the moisture concentration in the atmosphere. The dependence significantly (four times) decreased for four materials, including ABS White, as presented in Figure 7. However, on the other hand, there was 2–3 times the growth for PLA Lava and PET-G. The physical explanation of such behaviour requires additional research.

The dependence of the diffusion coefficient of ***w*_∞_** was approximated by the equation
$$D(w)=a_{D}{\left(w-w_{0}\right)}^{\beta }+D_{0}$$
where *a_D_*, *w*_0_, and *β* are parameters of the diffusion coefficient trendline presented in Table 3. For the materials with the independent diffusion coefficient *D* on ***w*_∞_**, its value is also given in the table.

The last evaluated parameter was the time taken for a sample to reach 80% of its equilibrium moisture content, *w*_80_, for the highest RH, which was 97% for long cylinders. The parameter was calculated using linear regression and is presented in Table 3. This parameter gives an approximate estimate of the time necessary to reach equilibrium for the filament in humid environments.

### 4.2. Swelling

The moisture sorption of all tested filaments was accompanied by its swelling, measured on “long cylinder” samples. The typical swelling process for Antero is illustrated in Figure 8.

The swelling process, in general, follows the moisture absorption process. The swelling strain ε is proportional to the moisture content *w* in the material, as seen from the example of Antero material in Figure 9.

Negative swelling strain means a shrinkage corresponding to moisture desorption from a material. The maximal swelling ability of the materials in equilibrium Δε (analogous to the moisture sorption capacity) is presented in Figure 10.

The relationship between equilibrium moisture content and swelling strain ε was approximated by the linear function
$$\varepsilon =a_{w}w_{\infty }+\varepsilon_{0}$$
where *a_w_* and ε_0_ are parameters presented in Table 4.

Most materials had a maximal swelling ability of 0.15−0.25%, which was not dependent on their structure and composition. Filled ABS ESD has a swelling ability significantly higher than that of ABS White. Nylon was an exception that shrank in humid environments by 2.43%.

### 4.3. Mechanical Tensile Tests

Mechanical tensile tests were performed on the samples moistened to saturation in the atmospheres mentioned above. This condition provides uniform moisture distribution across the sample section, which means an equilibrium state. Tension diagrams for four samples of ABS ESD moistened in an atmosphere with RH = 16%, as an example, are presented in Figure 11. The figure shows very little scatter in the data.

The initial segments of tension diagrams for representative samples with different moisture contents are presented in Figure 12. Significant changes in the slope of the diagrams proportional to the elastic modulus and tensile strength with moisture content indicate a strong dependence of the material properties on the moisture content.

Examples of changes in the elastic modulus *E* and strength σ* with moisture content for the ABS ESD material are presented in Figure 13.

The changes in the properties of the materials were approximated using the linear functions
$$\begin{array}{l} E=a_{E}w+E_{0}\\ \sigma *=a_{\sigma }w+\sigma_{0} \end{array}$$
with the parameters given in Table 5. The values of *E*_0_ and σ_0_ are the parameters of approximations that correspond to the materials in the initial state as produced.

The maximal relative reduction in the properties of the moistened materials is presented in Figure 14. Elastic modulus and strength for ABS ESD are more moisture-sensitive than for pure ABS White. Filler particles likely provided the electrostatic discharge of this material, essentially changing its structure and affecting its sorption and mechanical properties. At the same time, conductive and colour-changing additives reinforced PLA Cnd and PLA LAVA. No evident dependence was revealed of the reduction in the properties of moistened materials on their crystallinity or hydrophilic/hydrophobic nature. The data for Nylon were excluded from the figure because the reduction in elastic modulus was 83% and in strength was 42% for this material, which confirms the degradation of the polymer’s structure with water absorption. Thus, significant values radically change the chart’s scale, and the reduction in properties for other materials is hardly distinguished.

Based on the presented data, all tested materials could be classified as having low (changes ca. 2–4% or less), moderate (5–9%), or high sensitivity to moisture (more than 10%) in terms of the reduction in mechanical properties. This reduction in the materials’ stiffness and strength with absorbed moisture from the atmosphere should be considered for responsible applications.

## 5. Conclusions

The experimental investigation and modelling of the water sorption process of 12 typical polymer filaments used for FFF were performed. The data necessary for predicting the effect of moisture on the mechanical properties of FFF structures were obtained.

The experimental investigation of the moisture sorption process of 12 typical filaments used for FFF was performed in humid atmospheres with an RH from 16 to 97% at room temperature. Materials with high moisture sorption capacity were revealed.The solution of Fick’s second equation for the cylinder was obtained in series form. The convergence of the equation and possible simplified expressions were analysed.Fick’s diffusion model was applied to all tested materials, and a set of sorption parameters were found. The obtained equilibrium moisture content of materials in the wide range of humid environments allowed their moisture sorption isotherms to be built. The isotherms were classified as linear according to Henry’s law for nine tested materials and nonlinear Type 3 (BET classification) for others. The moisture diffusivity in the tested materials dependent on the relative humidity of the atmosphere was evaluated. The diffusion coefficient was independent of the relative humidity of the atmosphere for six materials. It essentially decreased for four materials and grew for the other two.The swelling of the filaments in a wet environment was investigated experimentally. Swelling strain changed linearly with the moisture content of the materials and reached up to 0.5% for some of them. The material that demonstrated the most swelling was Nylon, with a maximal swelling strain of up to 2.5%.The possible degree of degradation of the mechanical properties of the filaments due to moisture absorption was estimated. Based on the presented data, all tested materials could be classified as having low (changes ca. 2–4% or less), moderate (5–9%), or high sensitivity to moisture (more than 10%) in terms of the reduction in mechanical properties. This reduction in the materials’ stiffness and strength with absorbed moisture from the atmosphere should be considered for responsible applications.The molecular and sub-molecular structure of the polymers could affect the moisture sorption properties and could be changed during the absorption process, but further targeted research is necessary. Filler particles caused the increase in water sorption ability and degradation of the polymer in the ABS ESD case, while for PLA LAVA and PLA Cnd, it did not change or even strengthened the material.

## Figures and Tables

**Figure 1 polymers-15-02600-f001:**
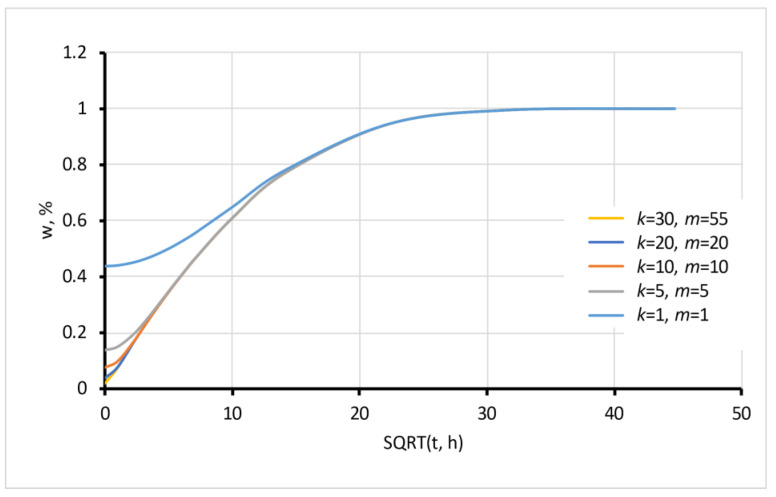
Modelled sorption with the different numbers of summands *k* and *m*.

**Figure 2 polymers-15-02600-f002:**
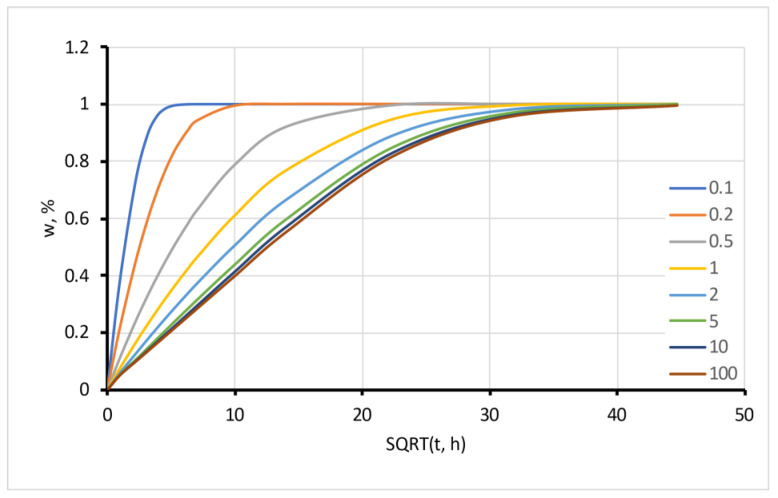
Modelled sorption using VBA add-in with different length-to-radius ratios, *l*/*R*.

**Figure 3 polymers-15-02600-f003:**
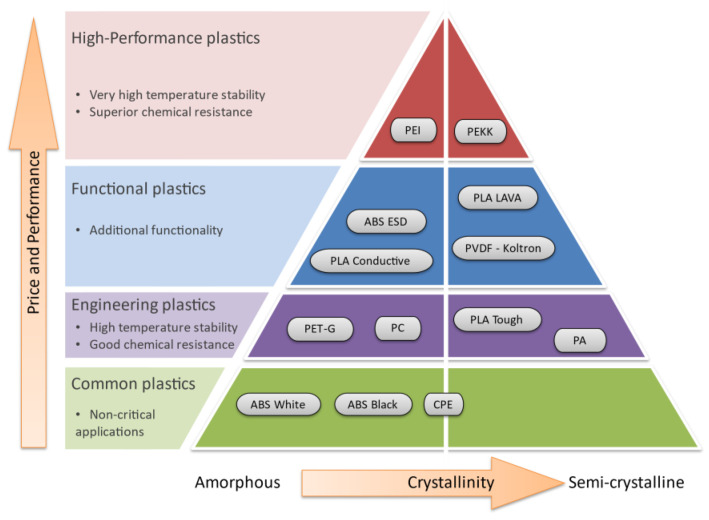
Categories of tested materials and their functionality.

**Figure 4 polymers-15-02600-f004:**
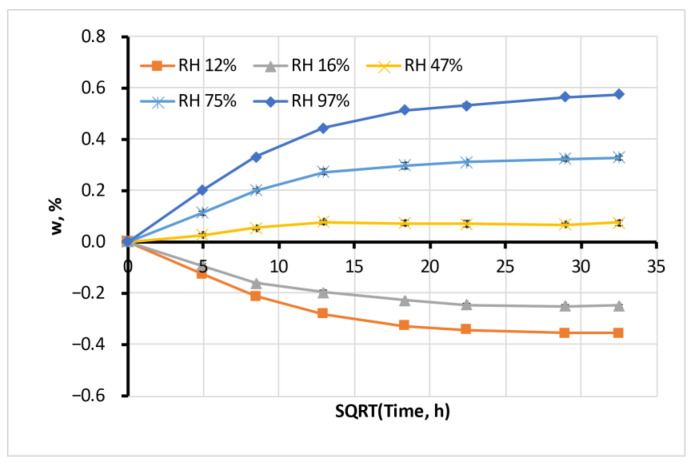
Average water sorption of Antero filament. Error bars are small and hardly visible.

**Figure 5 polymers-15-02600-f005:**
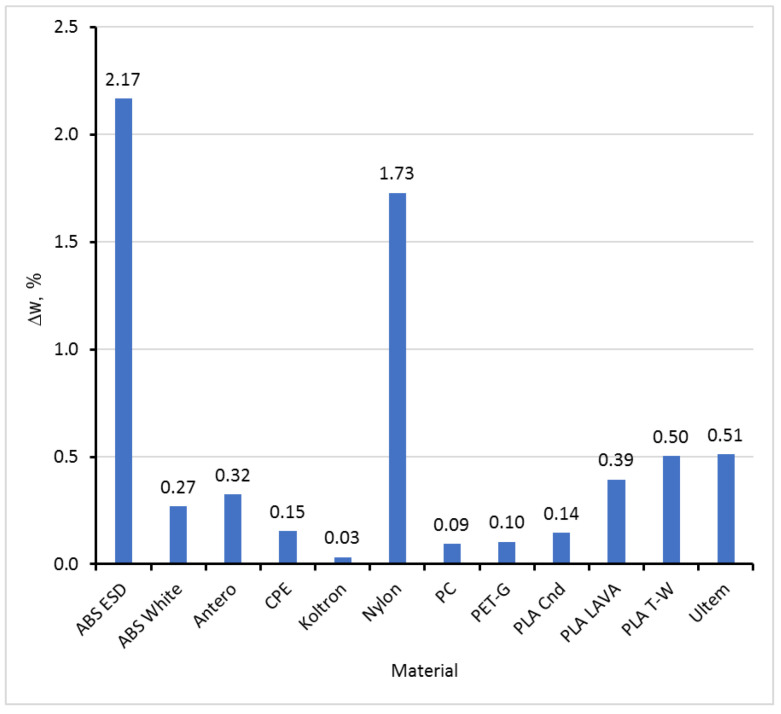
Maximal moisture sorption capacity of the tested materials, Δ*w*, in 24 h.

**Figure 6 polymers-15-02600-f006:**
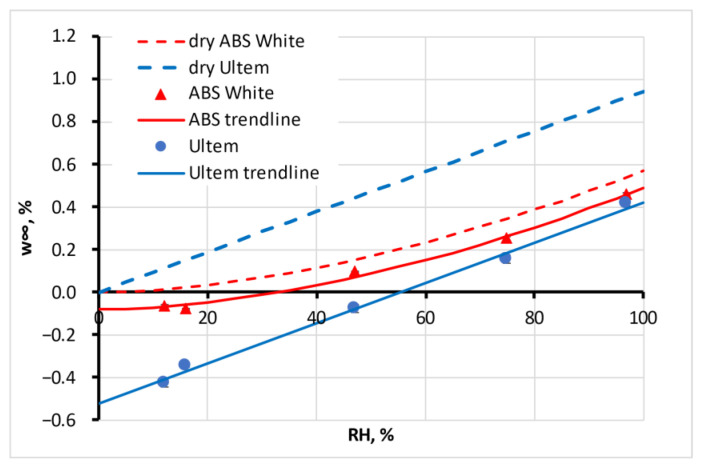
Sorption isotherm of Ultem (linear, Henry’s law, blue line, circles ●) and ABS White (BET, Type 3, red line, triangles ▲). Shifted approximations are for the materials if they should be initially dry (dashed lines). Error bars are small and hardly visible.

**Figure 7 polymers-15-02600-f007:**
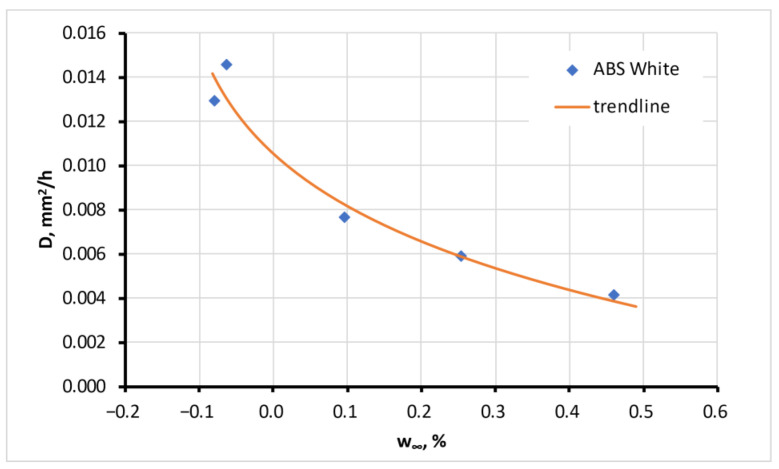
Diffusion coefficient of ABS White material depending on equilibrium moisture content, ***w*_∞_**. Error bars are small and hardly visible.

**Figure 8 polymers-15-02600-f008:**
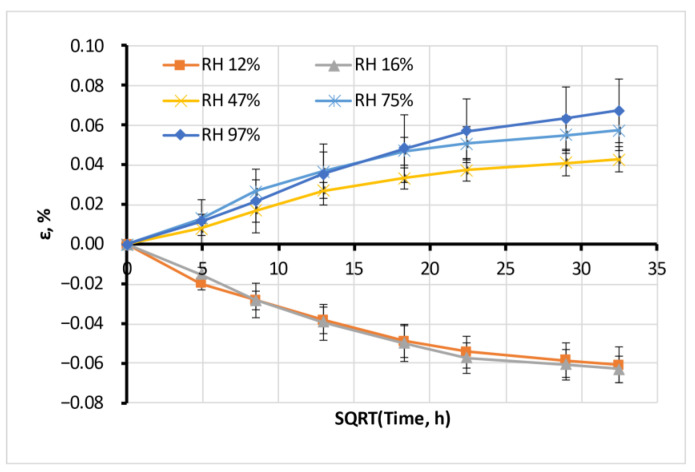
Average swelling of Antero material.

**Figure 9 polymers-15-02600-f009:**
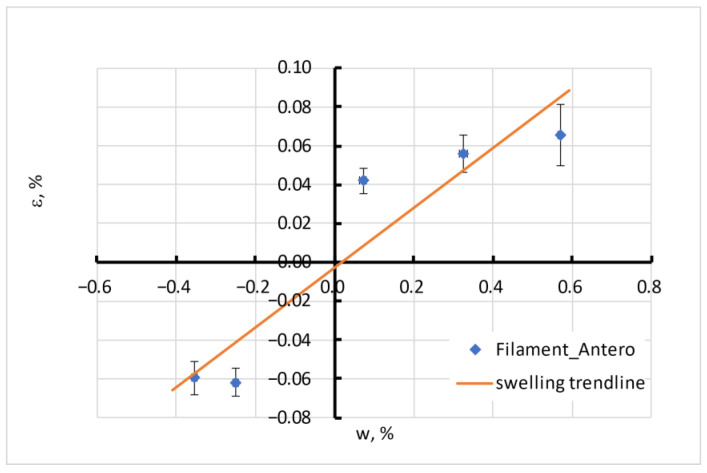
Equilibrium swelling of Antero.

**Figure 10 polymers-15-02600-f010:**
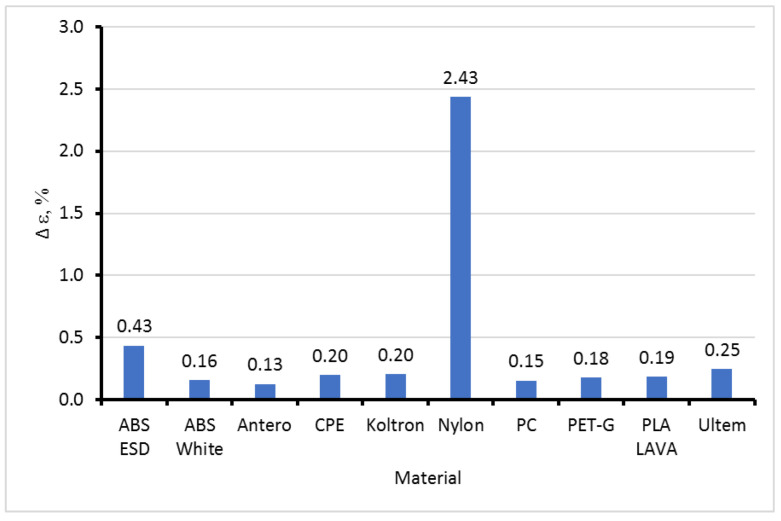
Maximal swelling ability (strain) of tested materials.

**Figure 11 polymers-15-02600-f011:**
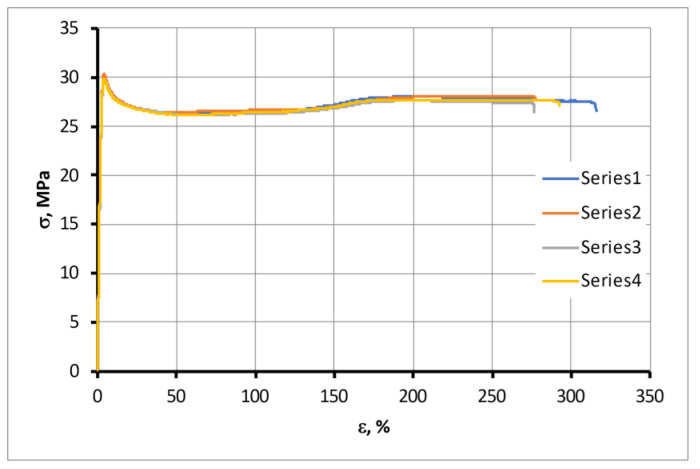
Tension diagrams for four ABS ESD samples moistened in an atmosphere with RH = 16%.

**Figure 12 polymers-15-02600-f012:**
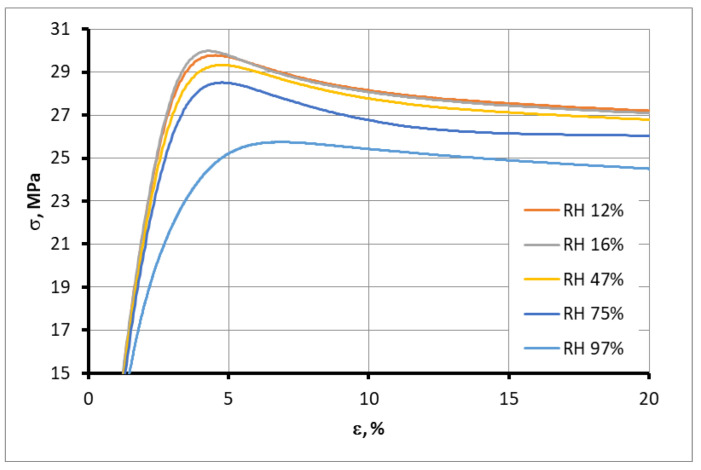
Initial segments of tension diagrams for representative samples of ABS ESD with different moisture contents.

**Figure 13 polymers-15-02600-f013:**
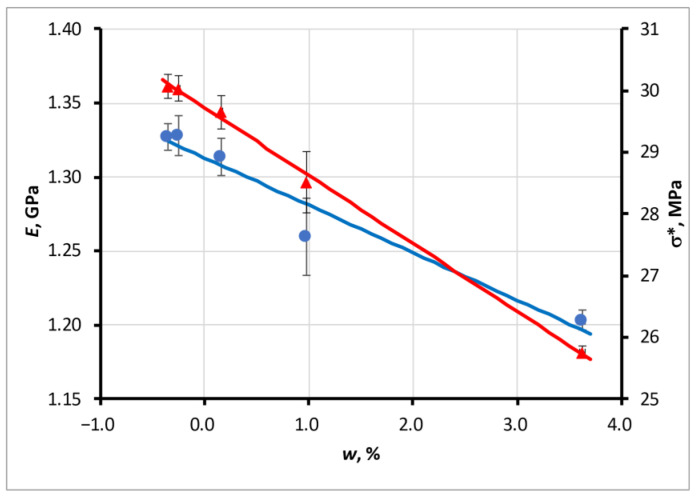
Example of changes in the elastic modulus *E* (blue line, circles ●) and strength σ* (red line, triangles ▲) with moisture content for the ABS ESD material.

**Figure 14 polymers-15-02600-f014:**
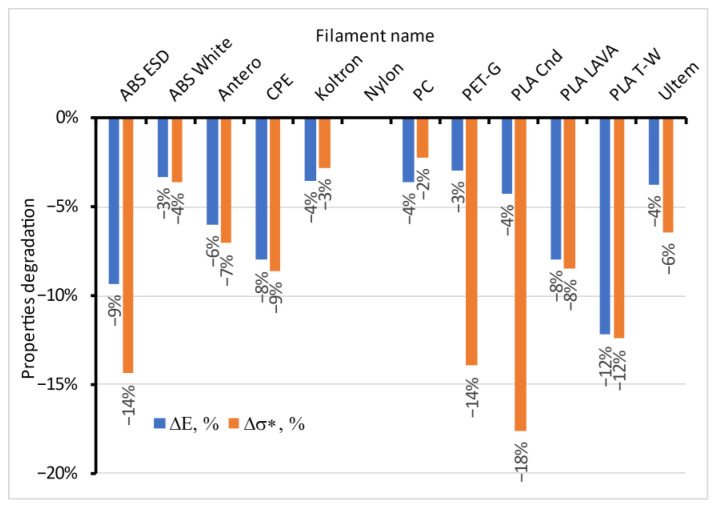
The maximal relative reduction in the elastic modulus and strength of the moistened materials.

**Table 1 polymers-15-02600-t001:** Filament types and notations used in this research (in alphabetic order).

Notation	Polymer	Industrial Name	Supplier	Density, kg/m^3^ [28]
ABS Black *	Acrylonitrile butadiene styrene	Ultimaker ABS (black)	Ultimaker, Utrecht, Netherlands	1124 ± 1
ABS ESD *	Acrylonitrile butadiene styrene	ABS-ESD	KIMYA, Nantes, France	1060 ± 2
ABS White	Acrylonitrile butadiene styrene	Ultimaker ABS (white)	Ultimaker, Utrecht, Netherlands	1124 ± 1
Antero	Polyetherketoneketone	Antero 800NA	Stratasys, MN, USA	1199 ± 3
CPE	Chlorinated polyethylene	Ultimaker CPE (white)	Ultimaker, Utrecht, Netherlands	1273 ± 2
Koltron *	Polyvinylidene fluoride	Koltron G1	Add:North, Fagersta, Sweden	1750 ± 8
Nylon	Polyamide 6/66	Ultimaker Nylon	Ultimaker, Utrecht, Netherlands	1103 ± 6
PC	Polycarbonate	Ultimaker PC (white)	Ultimaker, Utrecht, Netherlands	1191 ± 1
PET-G	Polyethylene terephthalate glycol	PET g (white)	Devil Design, Mikołów, Poland	1276 ± 3
PLA Cnd	Polylactic acid	Conductive PLA	Protoplant, WA, USA	1212 ± 2
PLA LAVA	Polylactic acid	Tri Color Change-Lava	HELLO3D, Shenzhen, China	1246 ± 1
PLA T-W *	Polylactic acid	Ultimaker Tough PLA (white)	Ultimaker, Utrecht, Netherlands	1227 ± 2
Ultem	Polyetherimide	ULTEM™ 9085 Resin CG	Stratasys, MN, USA	1173 ± 1

* “Short” cylinder samples.

**Table 2 polymers-15-02600-t002:** Coefficients of the approximation (7) for all tested materials.

Filament Notation	Isotherm Type	Sorption Isotherm Coefficients			Aim Function	Δ*w*_∞_
		*a*	*b*	*c*		
ABS ESD	BET theory, type 3	4.145	−0.304	3.5	0.115	3.865
ABS White	BET theory, type 3	0.572	−0.082	1.76	0.012	0.157
Antero	Henry’s law	1	−0.408	1	0.021	0.925
CPE	Henry’s law	0.698	−0.212	1	0.001	0.519
Koltron	BET theory, type 3	0.126	−0.017	1.7	0.005	0.125
Nylon	BET theory, type 3	8.689	−0.503	2	0.23	8.127
PC White	Henry’s law	0.582	−0.242	1	0.033	0.55
PET-G	Henry’s law	0.632	−0.201	1	0.043	0.696
PLA Cnd	Henry’s law	0.898	−0.469	1	0.019	0.8
PLA LAVA	Henry’s law	0.817	−0.21	1	0.018	0.724
PLA T-W	Henry’s law	0.795	−0.378	1	0.42	0.706
Ultem	Henry’s law	0.944	−0.522	1	0.021	0.844

**Table 3 polymers-15-02600-t003:** Diffusivity and time to reach 80% moisture content of the materials.

Filament Notation	*D*, mm^2^/h	*a_D_*, (%)^−1^	*w*_0_, %	*β*	*D*_0_, mm^2^/h	Aim Function or Average Deviation Values	Time to Reach *w*_80_, h
ABS ESD		−0.9912	1.0000	0.0053	−0.4436	0.0013	178
ABS White		−0.066	0.068	0.077	−0.143	0.0007	101
Antero	0.00245					0.00011	181
CPE	0.00247					0.00003	165
Koltron		−0.0048	0.0033	0.3044	−0.0170	0.0001	550
Nylon	0.001					0.00060	360
PC	0.00472					0.00038	92
PET-G		0.00335	0.00128	1.00000	−0.20104	0.00011	125
PLA Cnd		−0.0049	0.0069	1.0000	−0.4695	0.0002	164
PLA LAVA		0.00782	0.00506	1.00000	−0.20966	0.00059	46
PLA T-W	0.0151					0.00052	27
Ultem	0.00399					0.00013	89

**Table 4 polymers-15-02600-t004:** Parameters of equilibrium swelling strain dependence on moisture content.

Filament Notation	*a_w_*, (%)^−1^	ε_0_, %	Aim Function
ABS ESD	0.108	0.003	0.067
ABS White	0.343	0.021	0.016
Antero	0.154	−0.003	0.017
CPE	0.365	−0.041	0.017
Koltron	1.787	−0.021	0.051
Nylon	−0.295	0.166	0.477
PC	0.308	−0.022	0.021
PET-G	0.295	0.001	0.029
PLA LAVA	0.285	−0.011	0.034
Ultem	0.402	0.033	0.05

**Table 5 polymers-15-02600-t005:** Mechanical properties of moistened filaments.

Filament Notation	*E,* GPa		σ*, MPa	
	*a_E_*	*E* _0_	*a_σ_*	*σ* _0_
ABS ESD	−0.032	1.313	−1.104	29.727
ABS White	−0.107	1.705	-1.665	39.589
Antero	0.010	2.774	−6.589	83.442
CPE	−0.123	1.774	−8.548	59.694
Koltron	−0.003	1.114	−5.031	25.015
Nylon	−0.285	2.377	−11.155	204.379
PC	−0.067	1.946	−2.537	63.190
PET-G	−0.085	1.967	−11.241	54.916
PLA Cnd	−0.073	1.962	−6.381	27.415
PLA LAVA	−0.241	2.716	−5.328	50.505
PLA T-W	−0.313	2.487	−7.840	45.815
Ultem	−0.028	2.309	−6.149	84.017

## Data Availability

Data sharing not applicable.

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
