# Peer review of "Moisture Sorption and Degradation of Polymer Filaments Used in 3D Printing"

_polymers, 2023, doi:10.3390/polym15122600_

Round 1
Reviewer 1 Report
The article is challenging to follow, as it presents two interrelated but distinct types of results. Half of the manuscript focuses on finding an analytical solution to the mass diffusion problem for a finite-length cylinder. While the separation method for three variables yields a mathematically interesting analytical solution, it significantly increases the reading complexity of the experimental part. It may not be suitable for publication in Polymers. The heavy mathematical writing of the experimental section is unlikely to engage Polymers' readership.
The simplistic transport model and the Dirichlet boundary condition do not consider sorption isotherms (except for small relative humidity steps), potential polymer swelling, physical (crystallization), chemical (hydrolysis) aging, or the diffusion coefficient's dependence on water content. Furthermore, it is still being determined why end effects play such a crucial role for filaments, which are, by definition, longer than wide. The analytical solution obtained through the separation of variables is mainly valid for long times. It is challenging to implement (involving first-kind Bessel functions) compared to mass-conserving numerical methods like the finite volume method. The authors do not provide enough terms for short-time implementation, nor do they propose a Laplace transform solution for short times.
Due to these reasons, I recommend splitting the paper into two separate publications. The experimental part should be revised and submitted to the journal Polymers. In contrast, the theoretical part should be published in an applied mathematics journal, where the solution's performance can be compared with existing solutions. For the Polymers article, the experimental results should be presented more conventionally:
- Sorption isotherms should be expressed in terms of dry mass.
- Hysteresis should be discussed.
- Potential recrystallizations or densification should be considered.
The diffusion coefficients' variation should be expressed as a function of water content, not relative humidity. Swelling or shrinkage should be evaluated. If a transfer model is proposed, it must account for the observed mechanisms.
At this stage, it is essential to divide the document into two separate manuscripts. This will help clarify the focus of each paper and better address the intended audience for each topic.
Author Response
Dear Reviewer 1,
Thank you very much for your positive comments on our paper. Your suggestions motivated us to read our paper once again with another accent. We agree that it was our fault to combine two different parts in one manuscript in the form it was in the first submission. The obtained analytical solution is useful for modelling moisture sorption by short cylindrical samples. On the other hand, in our opinion, the presented solution has not had such an outstanding scientific contribution to be published as a separate paper without verification using the presented results of the sorption experiments. We change the structure of the paper during the revision. We kept only Fick’s equation and the final analytical solution used for calculations in Sec. 3. All mathematical transformations we moved to the Sec. 7 Annex, which is supplementary. This restructuring was not highlighted in the revision because significant parts of the paper should be coloured.
We agree that the Dirichlet boundary condition does not cover many specific aspects like the physical and/or chemical ageing of the filaments, etc., mentioned by the Reviewer. On the other hand, Fick’s equation with the boundary condition by mathematical essence has not limitations for short-time sorption processes. The problem is the number of summands in the series to provide the necessary accuracy of the calculations. Zeros of Bessel function (30 values) are given in the Annex, Table 7 and provide enough accuracy (convergence of the series) even for short-time experiments. The obtained analytical solution is well applied for the description of specific moisture sorption processes performed in desiccators with constant RH and temperature with short cylinder samples that allow express evaluation of the diffusion coefficient and equilibrium moisture sorption values of the polymers. The obtained data will be used in the FEM simulation of moisture absorption by 3D-printed structures.
Sorption isotherms are expressed in terms of dry mass that correspond to the classical isotherm view, as presented in Figure 5 by dashed lines and text lines 237-240. This transformation was performed simply by setting b = 0 in (7) and Table 2. We also keep our original approximation by solid lines because it provides the 3D printing industry with information about the initial moisture content of delivered filaments. The changes are highlighted in yellow.
The main aim of the research was to perform an experimental investigation and model the water sorption process by typical polymer filaments used for FFF and get data necessary for further prediction of the moisture effect on the properties of FFF structures. It is clear that 13 tested polymers have different chemical formulations, molecular and sub-molecular structures and could change with moisture sorption. Recrystallisation and density of PLA filament and printed samples are discussed in:
Bute, I., Tarasovs, S., Vidinejevs, S., Vevere, L., Sevcenko, J., Aniskevich, A., “Thermal properties of 3D printed products from the most common polymers”, The International Journal of Advanced Manufacturing Technology. 124, 2739–2753, 2023. https://doi.org/10.1007/s00170-022-10657-7
The diffusion coefficients’ variation was expressed as a function of water content, not relative humidity. This was our mistake. Figure 6, Table 3, and text lines 262-269 were changed. The changes are highlighted in yellow.
Swelling and shrinkage were evaluated and discussed in Sec. 5.2 of the manuscript.
Reviewer 2 Report
The article deals with a very important applied topic for researching the properties of materials for 3D printing. The authors have done a great deal of experimental and theoretical work, built a number of models and relationships. But in this edition, a number of the results presented look weak: the journal has a polymer focus, therefore, the results obtained must be presented in the form of a comparison and the influence of the nature (structure) of polymer matrices on the process of water absorption and degradation of mechanical properties. In this presentation, the draft is very difficult to read and analyze. Thus, the authors should change the concept of presentation, discussion of results and conclusions.
I also add a few minor comments:
1) Give the conditions and methodology for sample preparation
2) How reproducible is the experiment on measuring the length of the samples (the authors write that there were difficulties in these measurements)? What was the experimental sample?
3) F3 - the authors write that undried samples were used to study water absorption. This will obviously lead to an error; in almost all standard methods for determining this parameter, pre-dried samples are used. The authors need to determine the initial water content.
Author Response
Dear Reviewer 2,
Thank you very much for your positive comments on our paper. We agree with your comment and have changed the concept of the presentation and the structure of the paper during the revision. We kept only Fick’s equation and the final analytical solution used for calculations in Sec. 3. All mathematical transformations we moved to the Sec. 7 Annex, which is supplementary. This restructuring was not highlighted in the revision because significant parts of the paper should be coloured.
Conditions and methodology for sample preparation were shortly mentioned in Sec. 4 Materials and methods. Text was added: All samples were cut by a knife and polished cylinder ends. No special procurement were applied.
The length of “long” cylinders (100 mm) was measured for swelling calculations. Mitutoyo Absolute digital micrometre 100-125 mm with a precision of ± 0.001 mm was used for the measurements. The problem was the filament sample of 100 mm in length taken from the spool kept an arc form and behaved as a spring. A lab-made fixture was used to keep the sample straight during the length measurements. Text lines 199-200: Multiple length measurements for “long” samples were performed to reduce the data scatter.
We intentionally started our sorption tests with the filament “as delivered” and not pre-dried because it provides the 3D printing industry with information about the initial moisture content of delivered filaments, even if those are kept in storage bags (most of filaments) or boxes (like industrial Antero and Ultem) with silica gel. The initial moisture content of the filaments was obtained during the experiments in the desiccators with a dry atmosphere. Moisture desorption curves are presented e.g. in Fig. 3 for Antero. The initial moisture content is parameter b given in Table 2 for all filaments.
Round 2
Reviewer 2 Report
The authors corrected minor comments, but the main ones remained unchanged. As I already wrote, the journal has a polymer theme and the article deals with filaments of various nature (in terms of composition and functional groups), so it is necessary to compare (correlate) properties (moisture absorption and degradation) with the chemical structure of filaments. In this version of the draft, the authors only present the results obtained without discussing the influence of the nature of polymers on properties.
Author Response
Discussion about polymers was extended in the paper. Information about the hydrophobic and hydrophilic properties of tested polymers was added in Table 1 and discussed in the text Sec. 4. Figure 3 also was added. Discussion about the sorption behaviour of the materials was added in Sec. 5.1 and about the degradation of the properties in Sec. 5.3. Conclusion 5 was added. All corrections are marked with a yellow marker.
Our paper was intentionally targeted on “Polymers - Special Issue - Mechanical and Physical Properties of 3D Printed Polymer Materials” with special interests in mechanical engineering, materials engineering, …environmental effects, as mentioned in the call description by the guest editor.
The extensive set of experimental data given in the paper is necessary as input data for further analytical and FEM modelling of the long-term behaviour of 3D printed structures in engineering applications.
